# *CCL2*, *CCR2* Gene Variants and CCL2, CCR2 Serum Levels Association with Age-Related Macular Degeneration

**DOI:** 10.3390/life12071038

**Published:** 2022-07-12

**Authors:** Gaile Gudauskiene, Alvita Vilkeviciute, Greta Gedvilaite, Rasa Liutkeviciene, Dalia Zaliuniene

**Affiliations:** 1Department of Ophthalmology, Medical Academy, Lithuanian University of Health Sciences, Eiveniu 2, LT-50161 Kaunas, Lithuania; rasa.liutkeviciene@lsmuni.lt (R.L.); dalia.zaliuniene@lsmuni.lt (D.Z.); 2Laboratory of Ophthalmology, Neuroscience Institute, Medical Academy, Lithuanian University of Health Sciences, Eiveniu 2, LT-50161 Kaunas, Lithuania; alvita.vilkeviciute@lsmuni.lt (A.V.); greta.gedvilaite@lsmuni.lt (G.G.)

**Keywords:** AMD, *CCL2* (rs1024611, rs4586, rs2857656), *CCR2* rs1799865, CCL2 and CCR2 serum level

## Abstract

Background: Age-related macular degeneration (AMD) is the most common cause of progressive and irreversible blindness in developed countries. Although the pathogenesis is not fully understood, AMD is a multifactorial pathology with an accumulation of inflammatory components and macrophages and a strong genetic predisposition. Our purpose was to investigate the association between early AMD and *CCL2* (rs1024611, rs4586, rs2857656) and *CCR2* (rs1799865) single nucleotide polymorphisms (SNPs) and CCL2, CCR2 serum levels in a Lithuanian population. Methods: The study included 310 patients with early AMD and 384 healthy subjects. Genotyping of *CCL2* rs1024611, rs4586, rs2857656, and *CCR2* rs1799865 was performed using a real-time polymerase chain reaction method, while CCL2 and CCR2 chemokines serum concentrations were analyzed using an enzyme-linked immunosorbent assay. Results: We found that the G allele at *CCL2* rs1024611 was more prevalent in the early AMD group than in controls (29.2% vs. 24.1%, *p* = 0.032). Similarly, the C allele in *CCL2* rs2857656 is more common in the early AMD group than in controls (29.2% vs. 24.2%, *p* = 0.037). Binomial logistic regression revealed that each G allele in rs1024611 was associated with 1.3-fold increased odds of developing early AMD under the additive model (OR = 1.322; 95% CI: 1.032–1.697, *p* = 0.027) as was each C allele in rs2857656 under the additive model (OR = 1.314; 95% CI: 1.025–1.684, *p* = 0.031). Haplotype analysis revealed that the C-A-G haplotype of *CCL2* SNPs was associated with 35% decreased odds of early AMD development. Further analysis showed elevated CCL2 serum levels in the group with early AMD compared to controls (median (IQR): 1181.6 (522.6) pg/mL vs. 879.9 (494.4) pg/mL, *p* = 0.013); however, there were no differences between CCR2 serum levels within groups. Conclusions: We found the associations between minor alleles at *CCL2* rs1024611 and rs2857656, elevated CCL2 serum levels, and early AMD development.

## 1. Introduction

Age-related macular degeneration (AMD) is the leading cause of progressive and irreversible blindness in developed countries [1,2]. Early-stage AMD is more common than late-stage pathology [3] and includes drusen and retinal pigment epithelium (RPE) changes [4]. The prevalence of AMD has been reported for various ethnic groups [5,6,7] and accounts for 8.7% of all blindness worldwide [8]. As the ageing of society is an inevitable trend, AMD is expected to affect approximately 300 million people worldwide by 2040 [9]. 

Although the pathogenesis is not fully understood, AMD is certainly triggered by advanced age, gender, ethnicity, cigarette smoking, imbalanced diet, oxidative damage, genetic loci, and local chronic inflammation [4,10,11,12]. Photoreceptors are gradually affected by abnormal function of the RPE, and imbalance of oxygen and nutrient transport between the outer retina and vessels; in wet and advanced dry AMD, the permeability of choriocapillaris is also increased [13]. Drusen are undigested subretinal deposits composed of proteins, cholesterol, and oxidized lipoproteins, suggesting the role of oxidative stress in the pathologic process [4]. In addition, these deposits also involve components of the complement system and immunoglobulins [14]. Chemoattractant-rich drusen can trigger a low-grade inflammatory response and induce the recruitment of pro-angiogenic macrophages that can cause an AMD process [15].

The immune status also significantly impacts the AMD process [16]. Inflammatory cells detected in AMD lesions confirm the immune response aspect [17]. Several combinations have been investigated in the context of AMD; however, the most promising biomarker candidates belong to the complement system, lipid metabolism, and oxidative stress pathway [18,19]. Chemokine (C-C motif) ligand 2 (CCL2) is a member of the chemokine family responsible for monocyte chemotaxis and is encoded on chromosome 17 (chr.17, q11.2) [20]. CCL2 is produced by various cells, including epithelial cells, endothelial cells, microglial cells, and fibroblasts [21]. However, the primary origin is monocytes and macrophages [22]. CCL2, in principle, binds to the C-C chemokine receptor 2 (CCR2) encoded by the *CCR2* gene on chromosome 3 (chr.3, p21.31) [23]. However, other receptors, including CCR4, can also be involved in signaling cascade activation [24]. CCL2/CCR2 causes a cascade of signaling pathways associated with inflammatory, oncological, and atherosclerotic diseases [25,26,27,28]. Furthermore, CCR2 is a receptor that binds not only CCL2 but also other comparable chemokines CCL7, CCL8, and CCL13. This universality in chemokine-receptor interaction may lead to a similar or opposite effect depending on a particular pair [29]. RPE cells also generate CCL2 induced by inflammation, suggesting that RPE cells can cause macrophage accumulation in the subretinal area and choroidal tissue from circulating monocytes. This macrophage recruitment leads to increased secretion of pro-inflammatory substances and results in AMD [19], suggesting that inhibition of the CCL2/CCR2 signaling cascade may play a critical role in macrophage-triggered photoreceptor degeneration [30].

CCL2 levels and their association with AMD are controversial. Experimental studies have examined the increase in the secretion of the chemokine CCL2 in human and mouse RPE cells due to oxidative stress [31]. Other authors also found a significant association between higher CCL2 chemokine levels and AMD [19,31], while other studies found no significant change in levels in AMD compared to the control group [32,33,34].

In addition, AMD relation to *CCL2* and *CCR2* gene polymorphisms has been evaluated in clinical investigations with controversial results. Some studies published the association between *CCL2* and *CCR2* gene polymorphisms and AMD pathogenesis [19,35,36] and identified complement cascade genes, cytokines, and chemokine signaling pathways, including CCL2, as potential novel moderators of AMD [37]. In contrast, other studies have not confirmed the influence of these genes on AMD development [36,38].

Considering that the chemokine signaling pathway might be involved in the pathological processes leading to AMD pathogenesis, we included three single nucleotide polymorphisms (SNPs) in *CCL2* gene (rs1024611, rs4586, rs2857656) and one in *CCR2* gene (rs1799865) in our study.

Anand et al. investigated that individuals with both the exonic *CCL2* rs4586 and the *CCR2* rs1799865 SNP are at increased risk for AMD progression [19]. However, no association between the six SNPs in *CCR2* and five SNPs in *CCL2* and AMD has been demonstrated in the Caucasian population [38]. Another SNP in *CCL2* (rs1024611), located in the promoter region responsible for the transcription factor binding site and protein expression, has also been studied by several authors. Sharma and co-authors analyzed this SNP and claimed that both *CCL2* rs1024611 AG and GG genotypes are associated with AMD [35]. In contrast, other researchers found no significant differences in *CCL2* rs1024611 genotypes and allele frequencies between healthy individuals and AMD patients [39].

The final promoter SNP in *CCL2* (rs2857656 CC) was associated with a carotid plaque in African Americans from families with premature coronary artery disease [40]. In contrast, there was no significant association between *CCL2* rs2857656 and ischaemic stroke in the Korean population [41].

To our knowledge, the *CCL2* rs2857656 gene polymorphism was studied for the first time in AMD patients. The literature review suggests that the associations between these genes, chemokine level, and AMD may vary based on different ethnic groups. Therefore, we selected three SNPs in the *CCL2* gene (rs1024611, rs4586, rs2857656) and one in the *CCR2* gene (rs1799865) as possible genetic markers for early AMD. Our research aimed to determine the serum levels of CCL2 and CCR2 and identify the role of previous genes in early AMD in the Lithuanian unit.

## 2. Materials and Methods

### 2.1. Subjects

The research was conducted in the Department of Ophthalmology of the Lithuanian University of Health Sciences (LUHS) Hospital, Kaunas Clinics, and the Laboratory of Ophthalmology of the Neuroscience Institute of LUHS in accordance with the requirements of the Declaration of Helsinki. The Kaunas Regional Biomedical Research Ethics Committee approved the study (approval numbers: 9 July 2015 No. BE-2-26 and 26 January 2017 No. P1-BE-2-26/2015). Written informed consent was obtained from all subjects studied. Our study included 694 samples from 310 patients with early AMD in at least one eye and 384 healthy subjects without the ophthalmic disease.

### 2.2. Ophthalmological Evaluation

Complete ophthalmologic examination was performed in all subjects by an experienced ophthalmologist, including slit-lamp biomicroscopy and dilated ophthalmoscopy with tropicamide 1%. Swept-source optical coherence tomography (SS-OCT) was performed. AMD was classified based on the Age-Related Eye Disease Study (AREDS). An early AMD diagnosis was defined by multiple tiny drusen or a few intermediate drusen (63–124 μm in diameter) or changes in RPE (Figure 1). Early AMD was also differentiated from polypoidal choroidal vasculopathy as it is an exudative maculopathy with subretinal hemorrhage, pigment epithelial detachment, and neurosensory detachment, more common in non-white populations. Fluorescein angiograms were performed when necessary. Exclusion criteria included age less than 50 years, other ophthalmic diseases, and ocular abnormalities limiting fundus visualization.

### 2.3. DNA Extraction and Genotyping

The SNPs in our study were previously investigated in other populations for association with AMD or vascular disease and were selected because of their conflicting results. The *CCL2* and *CCR2* genes were selected for study because the chemokine CCL2 binds to the surface receptors of CCR2 and triggers chemotaxis of monocytes and basophils.

*CCL2* rs1024611, rs4586, rs2857656, and *CCR2* rs1799865 were analyzed in the Ophthalmology Laboratory at the LUHS Institute of Neuroscience. Deoxyribonucleic acid (DNA) extraction from white blood cells was performed by silica-membrane technology using a kit for genomic DNA extraction (GeneJET Genomic DNA Purification Kit, Thermo Fisher Scientific, Inc., Vilnius, Lithuania) according to the manufacturer’s protocol. 

SNPs were determined by the real-time polymerase chain reaction method (RT-PCR). All SNPs were determined according to the manufacturer’s recommendations using the Step One Plus RT-PCR Quantification System (Thermo Fisher Scientific, Inc., Singapore) using TaqMan^®^ genotyping assays (Thermo Fisher Scientific, Inc., Foster City, CA, USA). Assay IDs: rs1024611, C___2590362_20; rs4586, C__11939405_1; rs2857656, C____348241_10; rs1799865, C___2610509_30.

### 2.4. Genotyping Quality Control

Genotyping was controlled by selecting 5% of random samples for each SNP for repeated analysis, which confirmed absolute compliance rate of genotypes and alleles with the primary results.

### 2.5. Total Protein Estimation

Serum CCL2 and CCR2 concentrations were measured by enzymatic immunoassay (ELISA). This assay detects human CCL2 with a minimum detectable level of 2.3 pg/mL CCL2. Serum samples were diluted with assay buffer at a ratio of 1:5. ELISA was performed as described in the instructions for the CCL2 ELISA kit (#BMS281, Thermo Fisher Scientific Inc., Viena, Austria). Human CCR2 assay minimum detectable level—0.156 ng/mL CCR2, and the maximum 10 ng/mL. Serum samples were diluted with assay buffer at a ratio of 1:5. ELISA was performed as described in the instructions for the CCR2 ELISA kit (abx570562, Abbexa LTD, Cambridge, UK).

Absorbance was measured at 450 nm in a microplate reader (Multiskan Fc, Thermo Fisher Scientific Inc., Shanghai, China). A linear model was used to generate the standard curve, and the results were obtained after multiplication by the dilution factor (5×).

### 2.6. Statistical Analysis

Statistical analysis was performed with the SPSS 27 program (IBM SPSS, Armonk, NY, USA). Results are presented as absolute numbers with percentages in parentheses and median and interquartile range (IQR). Serum CCL2, CCR2 levels, and age between study groups were compared using the Mann–Whitney U test. Comparisons of observed and expected frequencies of polymorphisms (rs1024611, rs4586, rs2857656, rs1799865) between groups were estimated with Hardy–Weinberg equilibrium using the χ^2^ test. The χ^2^ test was used to compare the distributions of *CCL2* rs1024611, rs4586, rs2857656, and *CCR2* rs1799865 SNPs in both groups. The risk of early AMD for the *CCL2* rs1024611, rs4586, rs2857656, and *CCR2* rs1799865 polymorphisms was predicted by logistic regression analysis expressed as an odds ratio (OR) with a 95% confidence interval (95% CI). Haplotype analysis was performed to evaluate the associations of haplotypes with early AMD using the online software SNPStats (https://www.snpstats.net/snpstats/) (accessed on 29 June 2022) [42]. The haplotype block was formed from three *CCL2* variants: rs1024611, rs4586, and rs2857656. Linkage disequilibrium analysis (LD) was assessed using measures D’ and r^2^. Associations between haplotypes are presented as ORs, 95% CI, and *p* values. Statistical significance was observed when the *p* value was <0.05.

## 3. Results

A total of 310 patients diagnosed with early AMD and age and gender-matched 384 healthy subjects were involved in the study (Table 1).

The distribution of genotypes of four SNPs (rs1024611, rs4586, rs2857656, and rs1799865) in the early AMD group and the control group showed no statistically significant differences between the groups. Otherwise, we found that the G allele in *CCL2* rs1024611 was statistically significantly more abundant in the early AMD group than in the control group (29.2% vs. 24.1%, *p* = 0.032) (Table 2). Similarly, the C allele in *CCL2* rs2857656 is statistically significantly more common in the early AMD group than in the control group (29.2% vs. 24.2%, *p* = 0.037) (Table 2). 

Binomial logistic regression revealed that each G allele at rs1024611 was associated with a 1.3-fold increased probability of early AMD development under the additive model (OR = 1.322; 95% CI: 1.032–1.697, *p* = 0.027), as was each C allele at rs2857656 under the additive model (OR = 1.314; 95% CI: 1.025–1.684, *p* = 0.031) (Table 3).

### 3.1. Haplotype Associations with AMD

We determined strong linkage disequilibrium between the studied *CCL2* SNPs (Table 4).

Haplotype analysis showed that haplotype T-A-G was most frequent and selected as a reference, and the C-A-G haplotype was associated with 35% decreased odds of early AMD development (OR = 0.65; 95% CI: 0.44–0.96, *p* = 0.031) (Table 5).

Considering the higher frequency of females in study groups, we also analyzed haplotype associations in female and male groups separately, but no statistically significant results were found (data not shown).

### 3.2. CCL2 Serum Levels

CCL2 serum levels were measured for 39 early AMD patients and 39 control subjects. Analysis showed elevated CCL2 serum levels in the early AMD group compared to control subjects (median (IQR): 1181.6 (522.6) pg/mL vs. 879.9 (494.4) pg/mL, *p* = 0.013) (Figure 2).

CCL2 serum levels were also analyzed by different genotypes of *CCL2* and *CCR2* polymorphisms, but no associations were found between genotypes and CCL2 serum levels.

### 3.3. CCR2 Serum Levels

CCR2 serum levels were measured for 39 early AMD patients and 39 control subjects. Analysis showed lower CCR2 serum levels in the early AMD group compared to control subjects; unfortunately, these differences did not reach statistical significance (median (IQR): 2.1179 (1.79) ng/mL vs. 2.3465 (1.30) ng/mL, *p* = 0.094) (Figure 3).

CCR2 serum levels were also analyzed by different genotypes of *CCL2* and *CCR2* polymorphisms, but no associations were found between genotypes and CCR2 serum levels (data not shown).

## 4. Discussion

AMD is a disease with multifactorial pathogenesis in which genetic predisposition also plays an important role. It is suggested that genetic variants in the genes responsible for the inflammatory response may influence the AMD process by altering the expression of chemokines.

The recent studies of 310 patients diagnosed with early AMD and 384 healthy controls may also indicate the importance of inflammation in AMD pathogenesis in the Lithuanian unit. We selected four SNPs (*CCL2* rs1024611, rs4586, rs2857656; *CCR2* rs1799865). *CCL2* rs2857656 was selected for the first time as a potential genetic risk factor for AMD pathology, whereas the others have been analyzed in the literature by different authors.

Our study showed that the haplotype of three *CCL2* SNPs was associated with a 35% decreased probability of early AMD development. In contrast, the distribution of genotypes of four SNPs (rs1024611, rs4586, rs2857656, and rs1799865) showed no differences between the groups. In parallel, Despriet’s study of two independent Caucasian populations from the Netherlands (357 cases and 173 controls) and the United States (368 patients and 368 controls) also showed no association between sequence variants in six *CCR2* SNPs and five *CCL2* SNPs and AMD [38]. Similarly, in the Iranian case-control study, which included 233 patients with advanced AMD and 159 healthy controls, the *CCL2* rs1024611 SNP was not associated with AMD [36]. However, Sharma and co-authors published different results in Indian subjects in which both *CCL2* rs1024611 AG and GG variants were associated with AMD pathogenesis [35]. Anand et al. also indicated that carrying both *CCL2* (rs4586) and *CCR2* (rs1799865) was associated with an increased risk of developing AMD in the same ethnic group [19], suggesting the importance of ethnicity in AMD pathology.

Analysis of *CCL2* rs2857656 found that the C allele at the *CCL2* promoter rs2857656 was significantly more frequent in early AMD. In addition, each C allele at rs2857656 was associated with an increased likelihood of early AMD occurrence. We found no study that examined the association between *CCL2* rs2857656 and AMD or other ophthalmic pathologies. However, data claim an association of *CCL2* SNP rs2857656 (−362GC) CC genotype with carotid artery plaques in the African American unit (*p* = 0.05) [40]. A South Korean study reached opposite conclusions, showing no association between *CCL2* rs2857656 and ischaemic stroke [41]. Although an association between vascular pathology and AMD is suspected, the results are still contradictory. Fernandez and coworkers found no association between the occurrence of stroke and AMD [43]. Other researchers found a relation between the development of AMD and an increased risk of myocardial infarction [44].

This study found that the G allele at the *CCL2* promoter polymorphism rs1024611 was more common in early AMD. Our study confirmed findings in the Chinese Han population of 129 AMD patients and 131 healthy volunteers, as they also found an increased incidence of AMD with the rs1024611 GG genotype and a significantly higher frequency of the rs1024611 G allele in AMD patients [45]. In addition, allele frequency analysis in the Indian population confirmed that the G allele was also more common in AMD patients than in the control group [35]. These data support the concept that the G allele stimulates the occurrence of AMD.

Our research tries to draw attention to the CCL2 chemokine and its receptor CCR2 in early AMD patients. Our analysis revealed higher serum CCL2 chemokine concentrations in early AMD patients. However, no correlations were found between serum CCL2 levels and the *CCL2* and *CCR2* polymorphisms genotypes. Several studies have analyzed chemokine levels. In a study of an Indian population, Anand et al. found higher serum CCL2 levels in AMD patients than in healthy individuals [19]. Zor and coworkers also confirmed the elevated CCL2 serum levels, but they analyzed patients with exudative AMD (eAMD) [46]. However, the findings of other authors were consistent with our results and reported no significant change in serum levels of this chemokine. Although Lechner and co-authors found increased CCL2 secretion in peripheral blood mononuclear cells due to oxidative injury in eAMD patients, there was no increase in serum CCL2 levels in these patients [32].

A study by Grunin et al. does not agree with our study; they explained similar serum levels of CCL2 in eAMD (286.1 ± 20.7 pg/mL) and control subjects (286.7 ± 51.4 pg/mL; *p* = 0.37) [33]. Falk published comparable results without significant change in plasma CCL2 levels [34]. Other authors analyzed the chemokine CCL2 in different ways. In experimental studies in mice, it was found that the CCR2/CCL2 pathway can be inhibited by suppressed macrophage infiltration [47]. It has been confirmed that genetically higher serum CCL2 concentration is associated with cardioembolic stroke [48]. Urinary biomarker analysis revealed a significant association between TGF-α1 and CCL2 levels in early AMD [49]. Examination of aqueous humor also revealed higher CCL2 levels than healthy controls and the association with advanced AMD (*p* = 0.03) [50]. Similarly, Jonas and coworkers published the increase of CCL2 concentration in aqueous humor (*p* = 0.07) and significant association with eAMD [51]. These results may highlight the importance of the interaction of CCL2 and its receptor in the immune response, leading to monocyte recruitment and macrophage infiltration in the damaged areas.

Our research revealed insignificantly lower CCR2 serum level in the early AMD group (2.3465 (1.30) ng/mL vs. 2.1179 (1.79) ng/mL, *p* = 0.094). Polish scientists determined plasma CCR2 levels in 100 patients with breast cancer (BC) and controls that involved 35 patients with benign breast tumors and 35 healthy women, as many cancer cells may express chemokines and chemokine receptors. Lubowicka et al. found that plasma concentration of CCR2 in the BC group and all stages of BC were significantly lower when compared to the healthy controls (in BC groups—0.96 (0.05–8.17), in the total control group—2.30 (0.05–22.60), ng/mL, *p* < 0.001 in all cases). They also confirmed differences between the concentration of CCL2 and CCR2 in individuals with benign breast tumors and healthy controls (*p* < 0.05) [52].

Although the interest in CCL2-CCR2 signaling was increased in recent years, mostly due to its relation to solid and metastatic cancers, CCR2 chemokine was not analyzed in AMD patients despite mediating not only a pro-tumorigenic function but also angiogenesis [53].

One of the limitations of our research was a relatively small sample size. Furthermore, the future target would be the inclusion of immune cell (monocyte/macrophage) analysis, chemokine concentration, and immune cell quantification in aqueous humor.

## 5. Conclusions

Immunogenetic factors affect early AMD development in the Lithuanian unit. Our study highlighted the associations between minor alleles in *CCL2* rs1024611 and rs2857656, elevated serum CCL2 levels, and early AMD development.

## Figures and Tables

**Figure 1 life-12-01038-f001:**
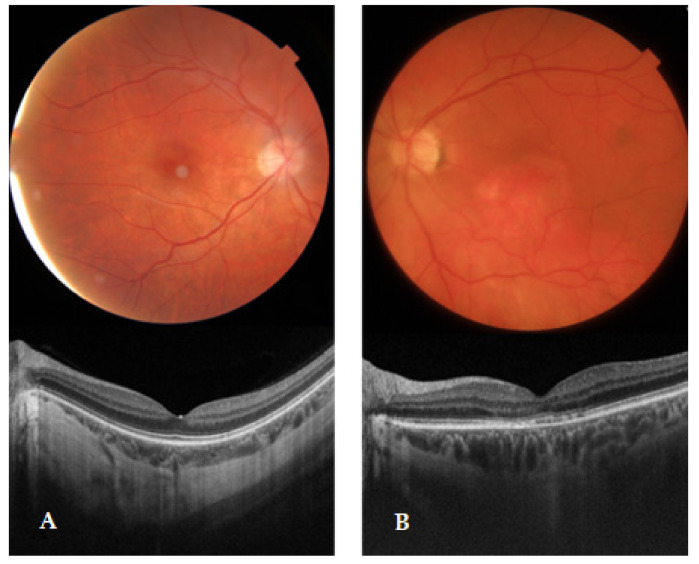
Fundus images and SS-OCT findings of normal (**A**) and early AMD (**B**) eyes. Patients were categorized according to the AREDS classification of AMD as follows: A—stage I, no or a few small drusen (<63 μm in diameter); B—stage II, multiple tiny drusen or a few intermediate drusen (63–124 μm in diameter) or changes in RPE. AMD, age-related macular degeneration; RPE, retinal pigment epithelium; SS-OCT, Swept-source optical coherence tomography.

**Figure 2 life-12-01038-f002:**
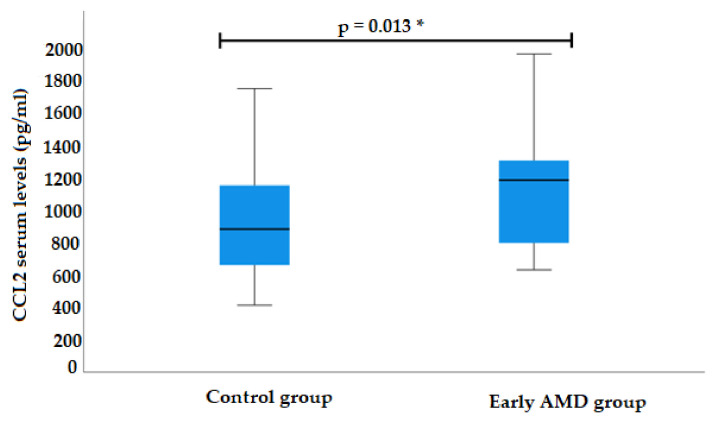
Serum levels of CCL2 in patients with early AMD and control group. AMD: age-related macular degeneration; CCL2: chemokine (C-C motif) ligand 2. * Significant values are presented as median and IQR.

**Figure 3 life-12-01038-f003:**
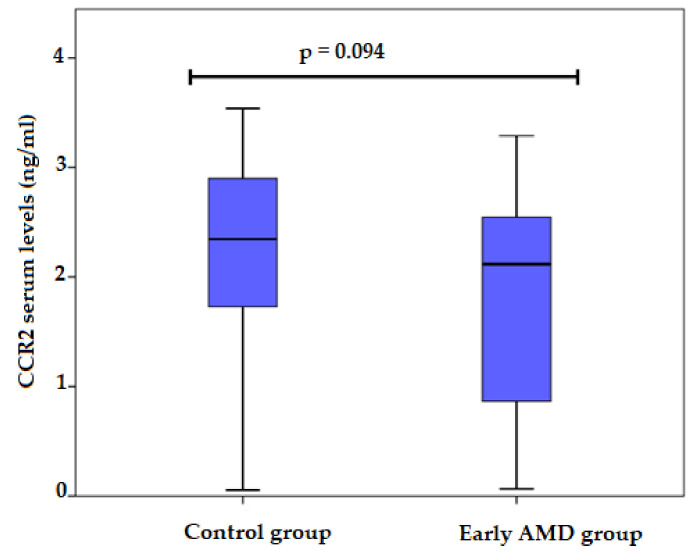
Serum levels of CCR2 in patients with early AMD and control group. AMD: age-related macular degeneration; CCR2: C-C chemokine receptor type 2. Values are presented as median and IQR.

**Table 1 life-12-01038-t001:** Characteristics of the study subjects.

Characteristic		Early AMD (*n* = 310)	Control (*n* = 384)	*p* Value
Gender, *n* (%)	Male	96 (31)	119 (31)	0.995
	Female	214 (69)	265 (69)	
Age (years), median (IQR)		76 (12)	75 (9)	0.097

AMD: age-related macular degeneration; IQR: interquartile range. Statistically significant when *p* value < 0.05.

**Table 2 life-12-01038-t002:** *CCL2* (rs4586, rs1024611 and rs2857656) and *CCR2* (rs1799865) SNPs in patients with early AMD and control groups.

Gene/Marker	Genotype/Allele	Early AMD, *n* (%)	Control Group, *n* (%)	*p* Value
** *CCR2* ** **rs1799865**	CC	26 (8.4)	26 (6.8)	0.723 0.577
	CT	121 (39)	152 (39.6)	
	TT	163 (52.6)	206 (53.6)	
	C	173 (27.9)	204 (26.6)	
	T	447 (72.1)	564 (73.4)	
** *CCL2* ** **rs4586**	CC	41 (13.2)	44 (11.5)	0.757 0.589
	CT	139 (44.8)	179 (46.6)	
	TT	130 (41.9)	161 (41.9)	
	C	221(35.6)	267 (34.8)	
	T	399 (64.4)	501 (65.2)	
** *CCL2* ** **rs1024611**	AA	152 (49)	216 (56.3)	0.078 0.032
	AG	135 (43.5)	151 (39.3)	
	GG	23 (7.4)	17 (4.4)	
	A	439 (70.8)	583 (75.9)	
	G	181 (29.2)	185 (24.1)	
** *CCL2* ** **rs2857656**	CC	23 (7.4)	17 (4.4)	0.087 0.037
	CG	135 (43.5)	152 (39.6)	
	GG	152 (49)	215 (56)	
	C	181 (29.2)	186 (24.2)	
	G	439 (70.8)	582 (75.8)	

AMD: age-related macular degeneration. *p* value < 0.05 indicated in bold is statistically significant.

**Table 3 life-12-01038-t003:** Binomial logistic regression analysis of the *CCL2* (rs1024611 and rs2857656) in patients with early AMD and controls.

SNP	Early AMD Model/Allele	OR; 95% CI; *p* Value
**rs1024611**	Additive G	1.323; 1.032–1.697; 0.027
**rs2857656**	Additive C	1.314; 1.025–1.684; 0.031

AMD: age-related macular degeneration. OR: odds ratio; 95% CI: 95% confidence interval. *p* value < 0.05 indicated in bold is statistically significant.

**Table 4 life-12-01038-t004:** Linkage disequilibrium between studied polymorphisms. Haplotype analysis.

	rs4586 (D′; r^2^)	rs1024611 (D′; r^2^)	rs2857656 (D′; r^2^)
rs4586 (D’; r^2^)	-	0.995; 0.654	0.991; 0.651
rs1024611 (D’; r^2^)	-	-	0.996; 0.989
rs2857656 (D’; r^2^)	-	-	-

**Table 5 life-12-01038-t005:** Haplotype association with the early AMD.

Haplotype	rs4586	rs1024611	rs2857656	Frequency AMD Group	Frequency Control Group	OR (95% CI)	*p* Value
1	T	A	G	0.64	0.65	1	-
2	C	A	G	0.066	0.106	0.65 (0.44–0.96)	0.031

AMD: age-related macular degeneration. OR: odds ratio; 95% CI: 95% confidence interval. *p* value < 0.05 indicated in bold is statistically significant.

## Data Availability

The data presented in this study are available on request from the corresponding author.

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
