# Peer review of "CCL2, CCR2 Gene Variants and CCL2, CCR2 Serum Levels Association with Age-Related Macular Degeneration"

_life, 2022, doi:10.3390/life12071038_

Round 1

Reviewer 1 Report

-        Overall, the manuscript is well structured and easy to understand

-        Although many similar studies have been carried out, this manuscript remains original because it was carried out on a population that has never been studied considering that genetic diversity is locally specific.

-        The introduction provides sufficient background and includes all relevant references, but it is too long. It would be better if it was presented more concisely.

-        All of the cited references are relevant to the research

-        The research design is appropriate. This study uses a large sample size, so it is hoped that the conclusions obtained will reflect the situation in the population. But the author does not explain how to separate AMD and PVC patients, because both have almost the same funduscopy and OCT features.

-        The methods are adequately described

-        The results are clearly presented

-        The conclusions are supported by the results

-        There are some typo errors, for example:

Line 23: rs102461

Line 200: rs102461

Line 184: … when p-value < was 0.05

-        English language and style are fine

Reviewer 2 Report

The authors have presented in the manuscript gene variants in CCL2 and CCR2 that are potentially associated with AMD. I found the paper interesting. I had a few issues with certain statements that I will outline below.

Line 15: gene should be nucleotide

Line 45: I would prefer a better term than genetic markers. Perhaps genetic loci or mutations or even polymorphisms.

Line 47: Please add a short bit in this sentence stating that the choriocapillary permeability is in wet and and advanced dry AMD.

Line 64: There should be a 2 after the word receptor.

Line 73: This sentence is confusing as written. The RPE cell cannot produce CCL2 in the choroid as the cells are not in the choroid. I am thinking you mean they produce it and it gets transported to the choroid. But that that is not how the sentence reads. Please reword.

Line 78: Is "macrophage-triggered macrophage photoreceptor degeneration" what is meant to be written here. It sounds awkward. Please reword.

Line 86: Please change the phrase "explained to AMD." The wording is confusing.

Line 125: Were the patients cleared of other ocular disease? The authors mentioned "ophthalmic disease limiting fundus visualization" but that is mainly diseases mostly affecting the anterior segment. 

Line 184: I believe you have your < and was reversed. It should say p value was <0.05.

Line 230: please add that the ccr2 levels were not significant.

Line 310: There appears to be an extra space between this paragraph and the next.

I also had some issues with experimental design and the data shown. I saw that the authors reported the sex of the patients studied and that more were female than male. As we know that AMD tends to affect females more than males, I feel the authors should at least show some effort to correlate these haplotype SNPs with sex of the patients. 

As for the data shown, I would like a figure with some representative images of fundus photos, OCT, etc of the patients used in the study (normal and AMD). I think this would add to the manuscript in more than one way as currently, it consists of nothing but tables and a couple of graphs and it would give the readers an idea of what the early AMD phenotype looks like compared to normals. Not everyone knows what these phenotypes appear as. My final issue is with the conclusions section. The issue no real conclusions were stated. Please add to this some and state what conclusions you drew.
